# Resolution of superluminal signalling in non-perturbative cavity quantum electrodynamics

Carlos Sánchez Muñoz [1], Franco Nori [1,2] & Simone De Liberato [3]

Recent technological developments have made it increasingly easy to access the non-perturbative regimes of cavity quantum electrodynamics known as ultrastrong or deep strong coupling, where the light–matter coupling becomes comparable to the bare modal frequencies. In this work, we address the adequacy of the broadly used single-mode cavity approximation to describe such regimes. We demonstrate that, in the non-perturbative light–matter coupling regimes, the single-mode models become unphysical, allowing for superluminal signalling. Moreover, considering the specific example of the quantum Rabi model, we show that the multi-mode description of the electromagnetic field, necessary to account for light propagation at finite speed, yields physical observables that differ radically from their single-mode counterparts already for moderate values of the coupling. Our multi-mode analysis also reveals phenomena of fundamental interest on the dynamics of the intracavity electric field, where a free photonic wavefront and a bound state of virtual photons are shown to coexist.

[1] Theoretical Quantum Physics Laboratory, RIKEN Cluster for Pioneering Research, Wako-shi 351-0198 Saitama, Japan. [2] Department of Physics, University of Michigan, Ann Arbor, MI 48109-1040, USA. [3] School of Physics and Astronomy, University of Southampton, Southampton SO17 1BJ, UK. Correspondence and requests for materials should be addressed to S.D.L. (email: s.de-liberato@soton.ac.uk)

Large light–matter couplings achievable in solid-state cavity quantum electrodynamics (QED) setups have allowed to enter non-perturbative regimes in which the interaction energy is a non-negligible fraction of the unperturbed excitation energies. Classified as ultrastrong coupling[1] or deep strong coupling[2] accordingly to whether the interaction energy is of the order of, or larger than, the bare ones, those regimes have been both achieved in different solid-state implementations[3–19].

From the theoretical side, the investigation of these non-perturbative regimes proceeded through the analysis of archetypical Hamiltonians, adapted to model different physical implementations and parameter regimes. The quantum Rabi model, describing a single two-level system (TLS) coupled to a single mode of the electromagnetic field, stands out as the simplest and the most iconic of them. Presently well understood for arbitrary values of the coupling[20], it has been successfully employed to model the first observation of strong coupling[21] and, with some tweaks, of deep strong coupling[16]. Its mathematical properties[22] and the possible implementations with synthetic models[23, 24] have also become object of interest.

To what extent any particular physical implementation is faithfully described by the quantum Rabi model depends largely upon how well it satisfies two assumptions: the emitter behaves effectively as a TLS, and only a single mode of the electromagnetic field significantly couples with it. The validity of the latter assumption is far from universal, and it has often been recognized that when the coupling is large enough to significantly hybridize the emitter with higher-lying photonic modes, those should be included in the Hamiltonian description[17, 25–32].

The first major result of this paper will be to show, exploiting a simple gedanken experiment, that, at least in the case of cavities with an harmonic multi-mode structure, there is actually an intrinsic problem in the description of a emitter-cavity system in terms of the single-mode quantum Rabi model, which becomes unphysical in the deep strong coupling regime since it allows for superluminal signalling. In order to better understand the practical relevance of such a problem, we will then perform a rigorous analysis of the multi-mode version of the quantum Rabi model, exploiting both numerical and analytical approaches. Such analysis will reveal that the failure to consider higher-lying photonic modes has a profound impact already in the ultrastrong coupling regime, that is, for values of the coupling nowadays routinely achieved in experiments. So far, such observations have mainly consisted of transmission experiments probing the low-energy spectrum of the system[33]. It is worth noticing that, in the kind of systems we are focussing on, one can obtain a low-energy spectrum of the single-mode description that does not differ greatly from the full, multi-mode case if one uses distinct fitting parameters. However, in contrast to these previous works, our analysis reveals that the different nature of the eigenstates and their degeneracy have critical consequences on the system dynamics.

## Results

**The problem of superluminal signalling.** We will focus most of our discussion on the simple physical system sketched in Fig. 1a: a perfect, one-dimensional cavity of length $L$ coupled to a single TLS of frequency $\omega_x$ placed at its centre. When only the coupling to the lowest mode of frequency $\omega_c = \pi c/L$ is considered, such a system is perfectly described by the standard Rabi Hamiltonian (we take hereafter $\hbar = 1$):

$$H_R = \frac{\omega_x}{2}\sigma_z + \omega_c a^\dagger a - ig\sigma_x(a - a^\dagger). \tag{1}$$

In order to see how this Hamiltonian allows for superluminal signalling when $g \simeq \omega_x, \omega_c$, let us consider the situation sketched in Fig. 1b, with an observer placed close to one of the mirrors and the system initialized in a factorized state, with the TLS either in its ground $|g\rangle$ or excited $|e\rangle$ energy level and the cavity field in its vacuum state. Such a configuration can be prepared performing only local operations on the TLS, i.e. by non-adiabatically switching on its coupling to the cavity[34, 35].

After a timescale $\tau_R \approx 2\pi g^{-1}$, the Hamiltonian in Eq. (1) will lead to an evolution of the cavity field, conditional on the initial state of the TLS. The cavity mode is delocalized along the cavity and the observer can thus, measuring the local field, acquire an information on the initial state of the TLS, placed at a distance $\frac{L}{2}$. Unless $\tau_R \gg \frac{L}{2c}$, the observer can thus measure the state of the TLS, placed at a distance $\frac{L}{2}$, in a time smaller than $\frac{L}{2c}$. The above inequality can be expressed in terms of coupling and bare frequencies as $\omega_c \gg g$, showing that the parameter regime in which superluminal signalling becomes possible coincides with the non-perturbative coupling regimes of cavity QED.

**Multi-mode quantum Rabi model.** In order to better understand the impact of the single-mode approximation, we will study the same model of Eq. (1) but now considering the full, real-space

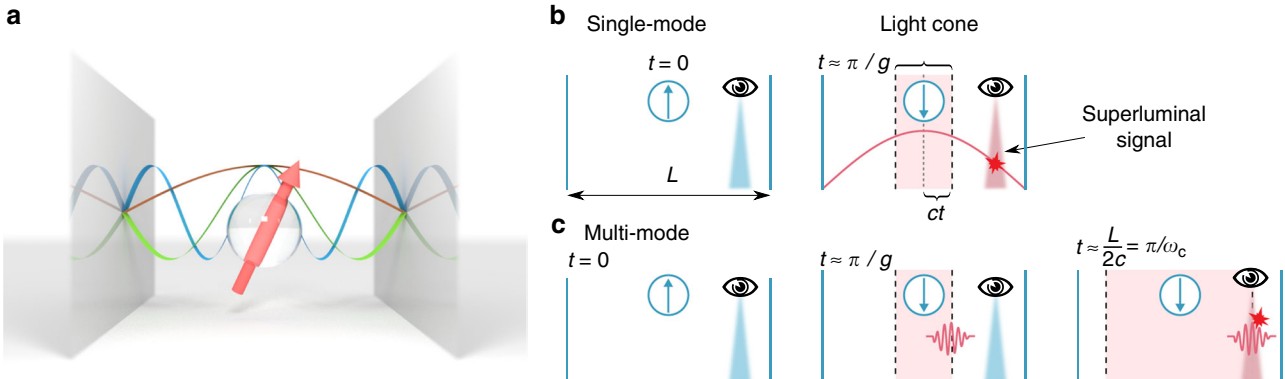

**Fig. 1** The problem of superluminal signalling in the single-mode Rabi model. **a** Schematic view of a qubit embedded in a perfect 1D cavity, together with the depiction of the three lowest cavity modes. When the qubit is only coupled to the fundamental mode, the system is described by the Rabi Hamiltonian. **b** Violation of relativistic causality by the single-mode Rabi model in regimes where $g \approx \omega_c$. An observer placed close to the cavity edge can retrieve information about the initial state of the TLS before light is able to reach its position. **c** A multi-mode description is able to capture the spatio-temporal structure of the light field necessary to comply with causality

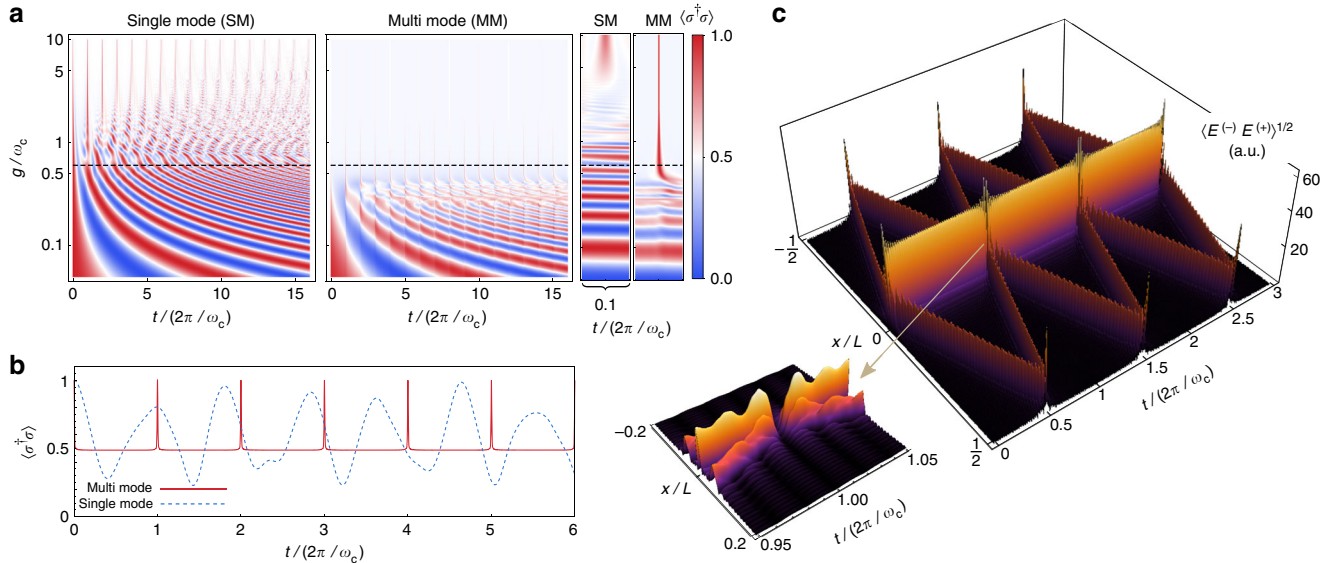

**Fig. 2** Breakdown of the Rabi model observed through the system dynamics. **a** Contour plot of the TLS population versus time and coupling rate. The dashed line marks the value $g/\omega_c \approx 0.6$ chosen for the rest of the simulations. Above this value, the single-mode Rabi model differs drastically from the multi-mode model. Insets on the right show a zoom view around a revival peak. **b** Population of an initially excited TLS versus time for the single-mode (blue, dashed) and multi-mode (red, solid) cases, for a coupling rate of $g/\omega_c = 0.6$. **c** Amplitude of the electric field inside the cavity (square root plotted for clarity) as a function of space and time, for $g/\omega_c = 0.6$. The inset focus on the precise moment when the field is perfectly absorbed by the emitter, giving rise to the revival peaks in the population of the TLS. Computed using the technique of MPS including 50 cavity modes

electric field inside the cavity:

$$\mathbf{E}(x) = i\mathbf{u}_z \sum_k \left(\frac{\hbar\omega_k}{2\epsilon_0 LA}\right)^{1/2} a_k e^{i(kx-\omega_k t)} + \text{h.c.}, \qquad (2)$$

where we have taken into account a single relevant polarization along the $z$ axis. Here $A$ is the transverse area of the cavity, and without any loss of generality, we have taken periodic boundary conditions to simplify the numerical analysis.

By defining the symmetric modes:

$$a_n = \frac{1}{\sqrt{2}}(a_k + a_{-k}), \text{ for } k = \frac{2\pi(n+1)}{2}, \ n = 0, 1, \ldots, \quad (3)$$

the dipolar coupling interaction $H_{\text{int}} = -\mathbf{d}\cdot\mathbf{E}$, where the dipole operator is $\mathbf{d} = \mu\sigma_x\mathbf{u}_z$, yields the multi-mode Rabi Hamiltonian:

$$H = \frac{\omega_x}{2}\sigma_z + \sum_{n=0}^{N-1}\left[(n+1)\omega_c a_n^\dagger a_n - i\sqrt{n+1}g\sigma_x(a_n - a_n^\dagger)\right], \quad (4)$$

with $g \equiv \sqrt{2\omega_c}\mu/\sqrt{2\epsilon_0 LA}$ and $N$ the total number of modes included in the description. Equation (4) is well defined in the electric dipolar approximation and the low-energy part of its spectrum converges in the limit of an ideal multi-mode cavity $N \to \infty$, when the TLS frequency $\omega_x$ includes the $N$-dependent renormalization due to the dipole self-interaction in the Power–Zienau–Woolley gauge[36, 37].

In the standard Coulomb gauge in which $\omega_x$ is microscopically independent from $N$, convergence would require instead to consider the diamagnetic $A^2$ term in the Hamiltonian[27, 28]. Recent works have proved that this remains true also in the case of superconducting circuits[31, 38, 39], assuring that our results are applicable also to this important class of systems. Given that we consider $\omega_x$ to be an experimentally measured value, we will not explicitly mark its dependency upon $N$.

In general, the total number of modes $N$ involved will depend on the specific physical implementation of the quantum Rabi model, e.g. due to the finite size of the emitter, with several tens of

them being a typical figure[31]. Even for these finite values of $N$, computing the dynamics of Eq. (4) for large $g/\omega_c$ is a computationally formidable task, because even in the ground-state each photonic mode contains a finite population of virtual photons[1]. As explained in the Methods section, we thus adopt the approach of refs. [40, 41], recasting the Hamiltonian into the form of a chain with nearest neighbour interactions, which can then be efficiently solved by using matrix product states (MPS)[42–44].

**System dynamics.** In Fig. 2a, we plot the time evolution of the TLS population versus $g/\omega_c$, with the TLS initially in its excited state and zero photons in the cavity, $|\psi(0)\rangle = |e\rangle|0\rangle$, obtained, respectively, solving Eq. (1) (single-mode) and Eq. (4) (multi-mode). This initial configuration is a superposition of excited states of the coupled light–matter system, which could be initialized by applying a $\pi$ pulse in a decoupled system and then by non-adiabatically switching on the coupling[34, 35]. As an alternative approach to obtain an initial excited configuration, one could also apply a suitable pulse to the coupled system in its ground state[45]. In any case, the effects that we report here appear as long as the system is initially in some superposition of excited states.

Figure 2b shows a plot along the dashed lines in Fig. 2a, corresponding to $g/\omega_c = 0.6$. It is clear that the single-mode approximation drastically fails as the system enters the non-perturbative region, with completely different physics taking place already for values of the coupling well below the boundary of the deep strong coupling regime. While for the considered values of the coupling the Rabi oscillations are distorted in the single-mode case, for the multi-mode Hamiltonian the TLS relaxes immediately and remains most of the time in a superposition of $|g\rangle$ and $|e\rangle$ yielding a population of $1/2$, experiencing a sequence of sharply peaked revivals that bring it back to the excited state at times multiple of the cavity roundtrip time, $2\pi/\omega_c$. Even for lower values of the coupling—before these revival peaks are fully formed—one can observe a perturbation of the Rabi oscillations taking place at those specific times. In Fig. 2c,

we plot the amplitude of the electric field inside the cavity, $x \in (-L/2, L/2)$ as a function of time, for the case $g/\omega_c = 0.6$. The electric field features the coexistence between two distinct components; (i): a localized cloud bound at the position of the TLS, and (ii): a free wavefront propagating at the speed of light. The free wavefront is backscattered at the edges of the cavity and returns at the position of the emitter at times $2\pi n/\omega_c$, when all the light is perfectly absorbed by the TLS—see inset of Fig. 2c— yielding the revival peaks in its population.

In order to gain further insight into the dynamical features of the multi-mode quantum Rabi model in the non-perturbative regime, we perform now an analysis similar to the one applied in ref. [2] to the single-mode case. To do so, we split the Hamiltonian into two parts, $H = H_I + H_{II}$, with $H_{II} = \frac{\omega_x}{2}\sigma_z$, and start by studying the action of $H_I$ alone. While in the single-mode case neglecting $H_{II}$ is a good approximation only in the limit $\omega_x \approx 0$ of the deep strong coupling regime[2], we will show that it is enough to describe the features that we have reported for the multi-mode model even at the resonant condition $\omega_x \approx \omega_c$ and in the ultrastrong coupling regime. Let us consider that $H_I$ is acting on a wavefunction whose matter component is one of the eigenstates of $\sigma_x$, $|\pm\rangle$. In that case, $H_I$ takes the form of a collection of driven harmonic oscillators:

$$H_{I,\pm} = \sum_{n=0}^{N-1} \left[ (n+1)\omega_c a_n^\dagger a_n \mp i\sqrt{n+1}\, g(a_n - a_n^\dagger) \right]. \quad (5)$$

The evolution under this Hamiltonian can be readily solved by means of a unitary transformation $U_\pm = \prod_n^{N-1} D_n\left(\frac{\mp\beta_0}{\sqrt{n+1}}\right)$, where $D_n(\beta) = \exp\left[\beta a_n^\dagger - \beta^* a_n\right]$ is a displacement operator acting on mode $n$ with a sign that depends on the state of the TLS, and $\beta_0 = ig/\omega_c$. This transformation gives a Hamiltonian without the driving term, $H_I' = U_\pm H_{I,\pm} U_\pm^\dagger = \sum_{n=0}^{N-1} \left[ (n+1)\omega_c a_n^\dagger a_n - g^2/\omega_c \right]$. We can write the evolution of an initial state with no photons $|\psi(0)\rangle_\pm = \prod_n |0\rangle_n |\pm\rangle$ under the effect of $H_I$ as:

$$|\psi(t)\rangle_\pm = U_\pm^\dagger e^{-iH_I' t} U_\pm |\psi(0)\rangle_\pm = e^{i\frac{g^2}{\omega_c^2}\sum_n^{N-1}\left\{1 - \frac{\sin[(n+1)\omega_c t]}{\omega_c(n+1)}\right\}} |\mp\xi_N(t)\rangle |\pm\rangle \quad (6)$$

where $|\mp\xi_N(t)\rangle \equiv \prod_n^{N-1} |\mp\beta_n(t)\rangle$. Here $|\beta_n(t)\rangle$ represents a coherent state in the $n$th cavity mode, with $\beta_n(t)$ given by:

$$\beta_n(t) = \frac{\beta_0}{\sqrt{n+1}}\left\{\exp[-i\omega_c(n+1)t] - 1\right\}. \quad (7)$$

The corresponding trajectories in phase space for each cavity mode are depicted in Fig. 3. The single-mode case was already introduced in ref. [2]; it features circular trajectories corresponding to oscillations around the centre of an harmonic oscillator displaced by $\beta_0$. The period of these oscillations is given by $2\pi/\omega_c$, and it is associated with the revivals in the probability of the initial state, corresponding to those times when the state in phase space crosses the $(0, 0)$ point. In the multi-mode case, this picture is extended, with each mode of frequency $\omega_c(n+1)$ following a circular trajectory, whose radius and period depend on $n$ as $1/\sqrt{n+1}$ and $1/(n+1)$, respectively. With high-energy modes oscillating faster than low-energy ones, the total period of the dynamics is fixed, as in the single-mode case, by the period of the fundamental mode, $\omega_c$. The revival probability of the initial state

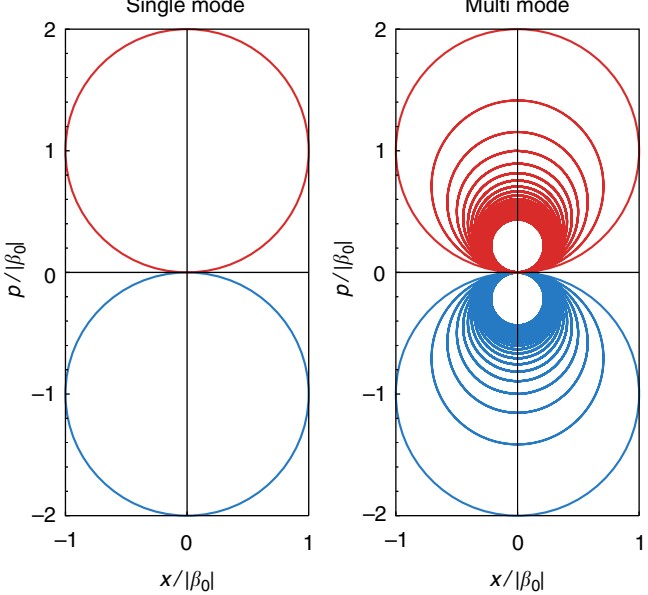

**Fig. 3** Phase space trajectories of the cavity modes. Trajectories in phase space for the single-mode case (left) and the multi-mode case (right) in which the trajectories of successive modes are plotted up to $n = 20$. Red (blue) curves correspond to the trajectories for an initial $|+\rangle$ ($|-\rangle$) state in the TLS

is given by:

$$P_0(t) = |\langle\psi(t)|\psi(0)\rangle|^2 = e^{-\sum_n^{N-1}|\beta_n(t)|^2}. \quad (8)$$

If the TLS is initially in an excited state $|e\rangle = (|+\rangle - |-\rangle)/\sqrt{2}$, as in the case we solved numerically, the resulting wavefunction consists of a superposition:

$$|\psi(t)\rangle = \frac{1}{\sqrt{2}}(|-\xi_N(t)\rangle|+\rangle - |\xi_N(t)\rangle|-\rangle), \quad (9)$$

with a revival probability given as well by Eq. (8). The two terms of the superposition are coupled by the Hamiltonian part $H_{II}$ that we have neglected so far, with a matrix element $\langle+|\langle-\xi_N(t)|H_{II}|-\rangle|\xi_N(t)\rangle = -\omega_x O_N(t)/2$ that is proportional to the overlap between the two cavity states, $O_N(t) \equiv \langle-\xi_N(t)|\xi_N(t)\rangle = e^{-2\sum_n^{N-1}|\beta_n(t)|^2}$. The exponent is given by a sum that diverges logarithmically with $N$ for all $t$ except for $t = 2\pi n/\omega_c$:

$$\sum_n^{N-1}|\beta_n(t)|^2 = \frac{g^2}{\omega_c^2}\sum_n^{N-1}\frac{2}{n+1}\left\{1 - \cos[(n+1)\omega_c t]\right\}. \quad (10)$$

This means that the overlap decays quickly to some stationary value $\overline{O}_N$ that goes to zero with increasing $N$ as $\overline{O}_N \approx 1/[2e^\gamma(N+1)]^{4g^2/\omega_c^2}$ (with $\gamma$ the Euler–Mascheroni constant) and then experiences sharp revivals at multiples of the cavity roundtrip time. In contrast to the single-mode case, where the width of the revival peaks is given by $g/\omega_c$, these decays and revivals occur on a short timescale $\tau \approx 2\pi/(N\omega_c)$, which justifies the approximation of neglecting $H_{II}$ as long as (i): the decay is fast enough, $N\omega_c \gg \omega_x$; and (ii): the stationary value of the overlap after the decay is small enough, $\omega_x \overline{O}_N \ll g$. This sets two conditions on $N$ and $g$ for the multi-mode physics to become

relevant and the effect of light propagation that we report to manifest, breaking down the single-mode Rabi physics. We have observed that, for $\omega_x = \omega_c$, values of $N \in [10, 100]$ and $g/\omega_c \gtrsim 0.25$ are sufficient to fulfil these conditions, meaning that these effects will be relevant already in the ultrastrong coupling regime for systems involving only several tens of cavity modes. A more detailed analysis of the implications of a finite $N$ is provided in Supplementary Notes 1 and 2. Interestingly, these results show that the multi-mode Rabi model can work as a dynamical description of wavefunction collapse based only on the Schrödinger equation. This is related to previous efforts[46–48], which, in the spirit of the many-worlds theory, describe the wavefunction reduction as a unitary evolution that includes the measurement device as part of the quantum system[49, 50].

As we showed before numerically, the revivals can also manifest in the population of the TLS, which within our approximation is trivially related to the overlap $O_N(t)$ as:

$$\langle \sigma^\dagger \sigma \rangle (t) = \frac{1}{2}[1 + O_N(t)]. \quad (11)$$

This expression reproduces perfectly the extremely sharp revival profiles that we report in Fig. 2b that were numerically computed for $N = 50$. Furthermore, it is easy to show how the collection of circular trajectories of the multi-mode case gives rise to the spatial profile of the electric field that we obtained numerically. The amplitude of the electric field is given by:

$$\langle E^- E^+ \rangle(x, t) = \frac{\hbar g^2}{\epsilon_0 A L \omega_c} \sum_{n,m=0}^{N} \left( e^{i(n+1)\omega_c t} - 1 \right)$$
$$\times \left( e^{-i(m+1)\omega_c t} - 1 \right) \cos\left[2\pi \tfrac{x}{L}(n+1)\right] \cos\left[2\pi \tfrac{x}{L}(m+1)\right], \quad (12)$$

which, when plotted, shows a perfect agreement to the profile in Fig. 2c. This is explicitly shown in Fig. 4a, which depicts a comparison between numerical calculations and Eq. (12) at a given time. Equation (12) can be decomposed into a time-dependent term, corresponding to (i) the part of the field that is emitted from the TLS and propagates freely towards the ends of the mirror, and (ii) a time-independent term, corresponding to the part of the field that remains bound to the TLS at the centre of the cavity. These terms have their origin in the time-dependent and -independent parts of the coherent amplitude $\beta_n(t)$ of each of the cavity modes, see Eq. (7), and the ratio between them will depend on the initial state (being 1/2 in our particular case).

**Propagative and bound photons**. The plot of the electric field in Fig. 2c seems to clearly attribute the regular peaks in Fig. 2a, b with period $2\pi/\omega_c$ to a rather trivial propagative effect of photons bouncing back and forth, and as such it had already been described in ref. [51] within the rotating wave approximation, which a priori excludes the presence of any non-perturbative effect. Still our analysis shows that those peaks have the same origin as those reported in ref. [2] for the single-mode quantum Rabi model in the deep strong coupling regime, in which, of course, the concept of propagation is non-relevant. Here we have shown that these two seemingly unrelated phenomena are effectively the same and that, in the multi-mode case, it is intimately related to light propagation and thus relativistic causality. This provides an intuitive physical understanding of why this phenomenon manifests at much lower coupling rates than actually predicted by the single-mode model: it is linked to a propagation that cannot be neglected when the coupling frequency becomes comparable to the cavity roundtrip, since it would allow for superluminal signalling.

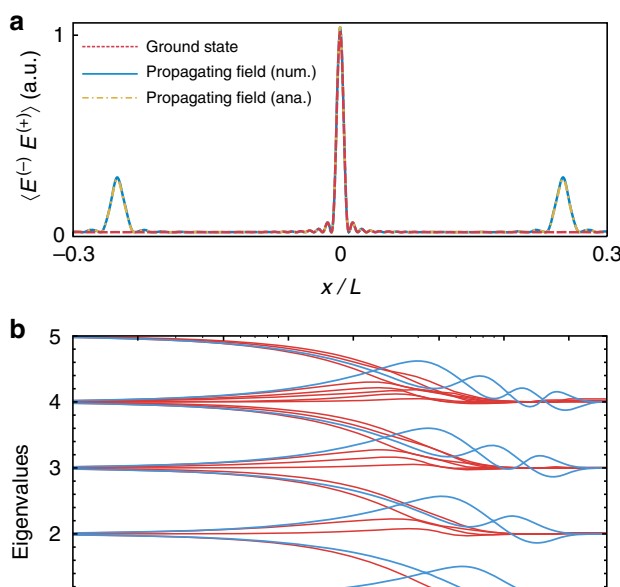

**Fig. 4** Ground state and eigenvalues of the multi-mode Rabi model. **a** Red, dashed line: Amplitude of the electric field inside the cavity corresponding to the ground state of the system for $g/\omega_c = 0.6$. Solid, blue (dashed-dotted, yellow): numerical (analytical) calculation of the electric field for an initial state $|e\rangle|0\rangle$ after evolving for a time $t = \pi/2\omega_c$, confirming that the dynamics of the system is given by the independent evolution of two freely propagating wavepackets plus a localized cloud of photons corresponding to the ground state of the light–matter system. **b** Low-energy spectrum of the single-mode (red) and multi-mode (blue) Rabi Hamiltonian as a function of the coupling rate. For each value of $g$, the eigenvalues are expressed with respect to the ground state. Here, $\omega_x = \omega_c$

In order to understand the second component of the dynamics (the localized cloud bound at the position of the TLS), let us recall that we expressed the Hamiltonian as a collection of displaced harmonic oscillators. Therefore, the absolute value of the time-independent part of $\beta_n(t)$ describes a coherent state at the equilibrium position of the $n$th displaced oscillator, $\beta_0/\sqrt{n}$, i.e. its vacuum state. We can then understand the time-independent part of the wavefunction as a set of displaced oscillators in vacuum, which corresponds to the ground state of the system. We have verified this by numerically computing the ground state using imaginary-time evolution, see Fig. 4a. The results obtained confirm that the ground state of a TLS non-perturbatively coupled to a cavity is indeed constituted by a localized cloud of photons around the TLS, which is in a superposition with a population corresponding to that observed in the revivals $\langle n_\sigma \rangle = 1/2$. Those virtual photons have been demonstrated to exist also in lossy systems[52], although once the coupling with the environment is properly considered[35, 53–56] their non-radiative nature becomes apparent. Our results provide a more transparent way to understand them as a localized, bound state of photons; in future works, the methods that we present here might be applied to study their properties in lossy systems. Bound states have already been documented in the context of ultrastrong coupling of a quantum emitter to open lines[57, 58], and there is much literature discussing their existence in boson impurity models in the single photon[59, 60] and, more relevant to our discussion, multiphoton case[61]. They are associated with eigenstates of the

system whose energy lie outside the energy spectrum of the bath, which in this case would be constituted by the infinite set of cavity modes.

For a given set of parameters, the spectrum of eigenvalues obtained from the multi-mode Rabi Hamiltonian strongly differs from the result given by the single-mode one, see Fig. 4b. In the large-coupling limit, both models feature a series of equispaced energy levels similar to the bare ones, a result shown above in the derivation of $H'_I$ and well known for the single-mode case[20, 27, 62, 63]. However, the results predicted by both models differ substantially in a range of couplings approximately delimited by $0.1 \lesssim g/\omega_c \lesssim 2$ for the low-energy eigenstates. The results shown in Fig. 4b evidence that transition energies should be fitted with a multi-mode Rabi Hamiltonian in order to obtain a proper description of the system; the use of a single-mode Rabi Hamiltonian might lead to a qualitatively similar prediction for the low-energy transitions but yielding an incorrect estimation of the system parameters. Owing to this possibility, an unambiguous evidence of the breakdown of the single-mode Rabi model physics enforced by causality should come from the analysis of the dynamics of observables, such as the TLS population, that, as we have shown, carry unequivocal signatures of the propagation of light inside the cavity.

## Discussion

We have performed a thorough theoretical analysis of a single emitter coupled to a photonic resonator. Our first result has been that, at least for resonators with harmonic spectra, like standard $\lambda/2$ cavities, the single-mode quantum Rabi model is incompatible with relativistic causality. By means of quasi-exact numerical calculations using MPS, we have then studied the multi-mode version of the quantum Rabi model confirming that, beyond certain values of the coupling rate, the single-mode model fails to describe the physics of a TLS coupled to the electric field inside a cavity. The failure of the model occurs in the regime of ultrastrong coupling, well before reaching the limit of deep strong coupling, and where the single-mode Rabi model is often invoked. This failure does not only manifest in the spectrum of eigenvalues, which differs from the one given by the single-mode model, but most importantly in the dynamics, which features freely propagating photonic wavepackets inside the cavity that coexist with a bound state of virtual photons corresponding to the ground state of the system.

Our theoretical analysis is most timely. Advances in superconducting circuits in fact not only recently led to the first observation of the deep strong coupling regime in a single-mode setup[16], but multi-mode effects in the ultrastrong coupling have also been recently reported[17]. Although this work primarily deals with the failure of the single-mode approximation, we verified that our results are not qualitatively affected by the breakdown of the TLS approximation. In Supplementary Note 3, we in fact extend our investigations beyond the quantum Rabi model, considering as matter degree of freedom a bosonic field with a small Kerr nonlinearity. We found that, in this situation, although higher modes are also involved in the dynamics, our conclusions remain valid.

These results bring a deeper understanding of a system of central importance in quantum mechanics and therefore are very relevant for the design of new technologies aiming to exploit the physics of light–matter coupling in the ultrastrong coupling regime.

## Methods

**Computation of system dynamics with MPS**. We make use of the approach presented in refs. [40, 41] and define a new set of operators by means of an unitary transformation $b_i = \sum_{n=0}^{N} U_{i,n} a_n$ to recast the Hamiltonian in Eq. (4) into another

with nearest neighbour interactions:

$$H = \frac{\omega_x}{2}\sigma_z + \sum_{i=0}^{N}\left[\omega_i b_i^\dagger b_i + t_i\left(b_i^\dagger b_{i+1} + \text{h.c.}\right)\right] - ig\rho_0\sigma_x\left(b_0 - b_0^\dagger\right), \quad (13)$$

with $U_{i,n} \equiv \sqrt{n+1}Q_i(n,1,0,N)\rho_i^{-1}$ ($Q_i$ being the Hahn polynomials); $t_i \equiv -A_i\rho_{i+1}/\rho_i$; $\omega_i \equiv 1 + A_i + C_i$ and

$$\rho_i^2 = \frac{(-1)^i(i+2)_{N+1}i!}{2(n+1)(-N)_iN!}, \quad (14)$$

$$A_i = \frac{(i+2)^2(N-i)}{2(i+1)(2i+3)}, \quad (15)$$

$$C_i = \frac{i^2(i+2+N)}{2(i+1)(2i+1)}. \quad (16)$$

where we used the Pochhammer symbol $(z)_i = z(z+1)\ldots(z+i-1)$. Writing the Hamiltonian in this form allows us to compute its dynamics very efficiently using the MPS method.

**Data availability**. The data that support the findings of this study are available from the corresponding author upon request.

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

## Acknowledgements

S.D.L. acknowledges support from EPSRC Grant No. EP/M003183/1. S.D.L. is a Royal Society Research Fellow. F.N. was partially supported by the MURI Center for Dynamic Magneto-Optics via the AFOSR Award No. FA9550-14-1-0040, the Army Research Office (ARO) under grant number 73315PH, the AOARD grant No. FA2386-18-1-4045, the IMPACT program of JST, the RIKEN-AIST Challenge Research Fund, the JSPS-RFBR grant No. 17-52-50023, the Sir John Templeton Foundation and CREST Grant No. JPMJCR1676. C.S.M. acknowledges support from a JSPS Postdoctoral Fellowship for Research in Japan (Short-Term) (PE17030).

## Author contributions

S.D.L. conceived the project. S.D.L. and C.S.M. developed the theoretical model. C.S.M. developed and performed the numerical calculations. S.D.L. and F.N. supervised the research. All authors discussed the results and its implications and contributed to writing the manuscript.

## Additional information

**Competing interests:** The authors declare no competing interests.

