## [Peer Review File · Nature Communications]

Reviewers' comments:

Reviewer #1 (Remarks to the Author):

In the paper "Multi-mode Quantum Rabi model and Superluminal Signalling", Sánchez-Muñoz et al compute the dynamics of a two level system inside a cavity. Their calculations are beyond the single mode approximation.

They argue that by increasing the light-matter coupling the single mode cavity description is not valid. They found that already at $g/w_c \sim 0.5$ the single and full mode descriptions yield qualitatively different results. In particular, they discuss that a single mode description gives an unphysical superluminal signaling.

I like the paper. I find it quite interesting and relevant in many physical implementations (specially in circuit QED). Clearly is rather fundamental and presents a big advance in the field. However, I think that for being published the writing must be improved and some points must be clarified. My main remarks are:

i) I am confusing with the superluminal signalling discussion. I am not an expert, thus, these discussions are always subtle for me. In QFT, causality imposes that two operators $A(x,t)$ and $B(y,t_0)$ acting on two space-like separated points (x,t) and (y,t_0) , must commute [Cf. Peskin and Schroeder QFT book as a textbook reference, Chap. 2 This is because causality means that local manipulations at (x, t) cannot influence in observables in (x', t') when both points are not causally connected.]. On the other hand, the propagator do not need to be zero outside the light-cone. In the ms. (last paragraph of the first page) the authors monitor the field dynamics at some point. However, they should discuss if these excitations are independent (or not) on local perturbations that are space like separated. They argue that they prepare the system in a factorized atom-cavity state. However, I do not see that this state can be prepared locally (e.g. only manipulating the atom) because it is a superposition of eigenvalues of the atom-cavity. The latter spread over the full cavity.

ii) In the introduction (3rd paragraph) the authors comment on the two main approximations in cavity QED, namely, the two level approximation and the single mode one. The latter is investigated in the ms. However, nothing is said on the TLS approximation. A naïve argument says that some of these modes can be in resonance with higher transitions in the atom. I understand that this is model dependent but, at least, it must be discussed what it is the role of the TLS approximation on their results.

iii) The authors cite several works [Their references 17, 23-29] where the multimode model was already discussed. Nature Comm. emphasizes novelty. Thus, they must clearly say what is novel in their work. What did the previous works discuss? Is the first time dealing with the dynamics?. Besides, two (at least) of these references report experiments. Are the effects described in the paper realizable in experiments?

iv) Below Eq. 14. They use only the acronym: "in the PZW gauge".

v) Below Eq. 14. What is the relation of w_x with its bare value?

vi) [Maybe related to point i)] The calculations are based on a very special initial condition. Do you have a protocol to prepare such a state in an experiment? What about other initial conditions?

vii) Their calculations use 50 modes [read in Fig. 2]. This introduces a cutoff of $50 w_c$. However, in a experiment the natural cutoff can be lower than $50 w_c$. What is the role of the cutoff (natural or numerical) in their results?

viii) For me it was surprising how localized are the wave packets travelling in the cavity. In principle, the atom is punctual and the emitting field should be a delocalized wave packet. Their calculations say that my guess is completely wrong, but I do not see where. Can the authors help me?

ix) The authors explain the field "cloud" around the TLS by computing the ground state. It would be also interesting to compute the $\langle \sigma_+ \sigma_- \rangle$ in the ground state to explain the $\frac{1}{2}$ between revivals [Cf. fig 2b]. They obtain this $\frac{1}{2}$ in Eq. (10) (without referring to Fig.2b). This analytical calculation was done in the limit $w_x=0$.

x) [Maybe related to viii)] I miss a physical picture explaining that adding modes the revivals and wave packet evolution become sharper.

xi) In the last figure (4b), the eigenvalues using the single mode and the multimode deviate already at $g=0.2$. Besides, the authors write at the end of the 5th paragraph, "... a low-energy spectrum of the single mode description that does not differ greatly from the full, multi-mode case, if one uses distinct fitting parameters...". If the experiments are fitted with a single mode, the coupling is over or under estimated?

Reviewer #2 (Remarks to the Author):

In their work, Munoz, Nori, and De Liberato discuss in depth the breakdown of the single-mode Rabi model as a description of cavity-QED once the regime of ultrastrong coupling is approached.

They first give an intuitive illustration of why the single-mode Rabi model cannot be valid in that regime, based on a Gedanken-experiment that would allow for superluminal signaling. Such a finding in an effective single-mode description is not surprising and occurs similarly in various different contexts, for example in trapped-ion quantum simulators addressing the center-of-mass phonon mode. Notwithstanding, it provides a nice, intuitive background. The authors then proceed to show numerically how the dynamics of the single-mode case drastically deviates from the multi-mode case. They furthermore give instructive analytical arguments employing a simplified model to explain their findings. The authors conclude by an analysis of the low-lying spectrum.

The study is very thorough and well-explained. It is also extremely timely in view of the recently strongly revived interest in the quantum Rabi model. It shows that a model that is commonly used to describe cavity QED has to be taken with great care when reaching the aspired regimes of ultrastrong and deep strong coupling, and it discusses the deviating physics that the more realistic multi-mode model presents. The article will thus be of high interest to many physicists and I recommend its publication in Nature Communications.

Before that, however, I would like the authors to address a couple of issues (many of them only cosmetic), which I outline below.

1. As indicated above, I find the strong focus on the superluminal signaling in title and abstract unnecessary and somewhat exaggerated, since it is not really surprising in such an effective single-mode approximation. Shifting the focus more strongly on the relevant parts of how a correct description is obtained and what its physical consequences are could improve the article. But this is maybe a matter of taste.

2. On p. 1, one finds “Unless $\tau_R \gg L/2c$. . .” Is not a simple “>” already sufficient?
3. In Eq. (4) and below, the symbol N is not properly introduced. Implicitly, it becomes clear that it is the number of modes, but that should be mentioned as well as why the limit $N \rightarrow \infty$ is taken.
4. Even if well-known, the acronym “PZW gauge” should be defined.
5. In Fig. 2c, the peaks above the dashed line are hardly visible, at least on my computer screen.
6. Again in Fig. 2c, in the multimode case there are vertical stripes, visible even at very low g . Are these numerical artifacts or are they physical?
7. Figure 4a shows the two contributions of the localized field and the propagating wave. What is the exact ratio between the amplitudes of both? From Eq. (12), derived analytic in the approximate model, it appears that it is a universal number, but in the numerics does it depend on the actual model parameters?
8. Around Fig. 4b, a discussion of the degeneracies of the levels at $g \rightarrow 0$ and $g \rightarrow \infty$ seems in order.
9. I am missing two references on experiments on the Rabi model:

1. Experimentally simulating the dynamics of quantum light and matter at deep-strong coupling

N. K. Langford, et al., Nature Communications 8, Article number: 1715 (2017)

2. Analog quantum simulation of the Rabi model in the ultra-strong coupling regime

J. Braumüller, et al., Nature Communications 8, Article number: 779 (2017)

These works realize the single-mode Rabi model in an effective way by alternating between Jaynes-Cummings and Anti-Jaynes-Cummings terms. In how far do the present findings apply to such synthetic models?

10. In the methods, the formula for the Pochhammer symbols in terms of a_i is confusing, since the symbols a_n are already used for the annihilation operators of the photon modes.

Reviewer #3 (Remarks to the Author):

In this work, the authors study theoretically the dynamics of a two-level system coupled to an optical resonator, focusing on the non-perturbative regimes of light-matter coupling, known as ultra-strong and deep-strong coupling, where the coupling rate is comparable to the resonator frequency. The authors show that, in those regimes, the single-mode description of the resonator is incompatible with relativistic causality and yields very different dynamics from the complete, multi-mode description of the field.

I think this is a really interesting and solid piece of work, reporting several important results. In particular, I believe that the argument of causality violation is specially relevant and timely, given the current state of the field, reporting ever-higher values of the coupling. The single-mode Rabi model has been used as a very standard tool to describe and analyze these systems in the ultra-strong coupling, and here the authors show unambiguously (see for instance the difference between single-mode and multi-mode behaviors in Fig. 1(a-b)) that something very important is missing in the single-mode description. I believe that it is most likely that works like the present one will help improving the quality and quantity of future publications in the field of cavity QED.

Also, the conclusions of the manuscript rely on quasi-exact calculations of the multi-mode Rabi model. To the best of my knowledge, this is the first time that such an exact analysis has been performed, which is an interesting result by itself, since it allows to explore all the regimes of light-matter coupling without resorting to any approximation. Such an analysis allows the authors to establish unambiguously what is the regime of light-matter coupling in which the single-mode description is no longer valid; notably this happens for values of the coupling rate lower to the best figures already achieved in the laboratory.

Despite I find the manuscript very complete overall, I have some comments I think should be addressed:

- Several experimental works (see for instance Bayer et al. Nano Lett., 17 , 6340 (2017), or Yoshihara Nat. Phys., 13, 44, (2017)) have already reported values of the light-matter coupling rate well above

$g/w = 0.6$, which is approximately where the authors show that the single-mode Rabi model is no longer valid. The authors should comment on the link between their work and these experimental results, and address to which extent are those results affected by the ones reported here.

- In this work, it is claimed that the well-known virtual photon population of the ground state in the ultra strong coupling regime (which is known to yield no emission) takes the form of a photonic cloud localized around the two-level system. I think the authors should comment on how does the spatial extension of this state depends on the coupling rate, and how could this observation be measured experimentally (given that these photons yield no observable emission).

- The color codes using in the manuscript are confusing sometimes. In particular, in Fig. 2(b), red is used to represent the multi-mode description, and blue for the single-mode. In Fig. 4(b), it is the other way around. I think this should be fixed for the sake of consistency.

As a conclusion, I find the manuscript interesting, with results that are both important and relevant in the current technological context. I therefore recommend publication in Nature Communications, provided my comments are addressed.

Reviewer #4 (Remarks to the Author):

In this work, authors compared two quantum Rabi models (QRM) of cavity QED. In ultrastrong coupling regime, considering only the lowest (or any single mode) will lead to an unphysical superluminal signal. Instead, the authors suggest that we should consider all possible modes of the cavity. They also gave numerical simulation and analytical approximated result to support this point.

After a brief research, some previous works mentioned that multimode should be considered when the coupling is strong enough, (such as Sundaresan, Neereja M., et al. "Beyond strong coupling in a multimode cavity." *Physical Review X* 5.2 (2015): 021035) but I did not find any article that showed single mode QRM would lead to superluminal signals. In my opinion, this paper has solid results and strong arguments while still needs to be revised and resubmitted. Following are some suggestions/questions and some mistakes I found.

Comments:

1. The title is somehow misleading. I think it should convey the idea that multi-mode QRM is better since it resolves the problem of unphysical superluminal signal that caused by single-mode QRM. Thinking about something like: "Resolution of Superluminal Signalling: Multi-mode Quantum Rabi Model".
2. I believe that the Hamiltonian (4) can be easily generalized to some other kinds of systems such as circuit-QED, where the coupling strength g_n to different modes are engineerable.
3. If only a finite number of modes are considered in equation (4), my naïve guess is that we can re-diagonalize the H matrix (4) to find only one superposition of photon modes coupled to the TLS, which might be able to reduce equation (4) back into equation (1).
4. Although the analytical result resembles the numerical solution, it might still need an analytical explanation: why is H_{II} neglectable compared with H_I ?
5. One main reason we are still using single-mode QRM is that driving and damping can be easily added in, can we extend it and add the driving and damping to this multi-mode picture?
6. The conclusion needs to be clearer: at what g/ω_c regime the single-mode QRM becomes invalid? (0.6? 0.2? or it depends?)
7. Is there a superluminal signal when coupling strength is small (not in ultrastrong regime)? If there is, why it is not a problem. If not, why it emerges after a certain threshold.

Typos:

1. In the captions of Fig. 2(c) and Fig. 4, I think ω_a should be ω_c .
2. In Equation (6), $|\psi(t)\rangle$ should be $|\psi(t)\rangle_{\rho m}$ as it represents two different states.
3. On page 4, ... seems to clearly attribute the regular peaks in Fig. 2 with period..., Fig. 2 should be specified as Fig. 2(b).

Wako-shi,
Southampton,
March 2, 2018.

Reviewer #1

I like the paper. I find it quite interesting and relevant in many physical implementations (specially in circuit QED). Clearly is rather fundamental and presents a big advance in the field. However, I think that for being published the writing must be improved and some points must be clarified. My main remarks are:

We thank the Reviewer for his/her time and consideration, and for the positive response.

1. i) I am confusing with the superluminal signalling discussion. I am not an expert, thus, these discussions are always subtle for me. In QFT, causality imposes that two operators $A(x,t)$ and $B(y,t_0)$ acting on two space-like separated points (x,t) and (y,t_0) , must commute [Cf. Peskin and Schroeder QFT book as a textbook reference, Chap. 2 This is because causality means that local manipulations at (x, t) cannot influence in observables in (x', t') when both points are not causally connected.]. On the other hand, the propagator do not need to be zero outside the light-cone. In the ms. (last paragraph of the first page) the authors monitor the field dynamics at some point. However, they should discuss if these excitations are independent (or not) on local perturbations that are space like separated. They argue that they prepare the system in a factorized atom-cavity state. However, I do not see that this state can be prepared locally (e.g., only manipulating the atom) because it is a superposition of eigenvalues of the atom-cavity. The latter spread over the full cavity.

Reply. We agree with the Referee that this is a subtle topic. Still, even without developing a full theory of perturbation propagation from a QFT perspective, we can answer affirmatively to the Referee's question: the initial state we chose, with the cavity in its vacuum state, can be prepared using only local operations on the qubit.

A simple protocol to prepare such a configuration can be implemented in a system in which the strength of the light-matter coupling can be non-adiabatically modulated. If the coupling is non-adiabatically switched-on at $t = 0$, and for $t < 0$ the uncoupled, empty cavity is in its ground state, at $t = 0^+$ the system will then be in a factorised state of the form we consider as initial condition. The condition for non-adiabaticity, that is the timescale on which the coupling has to be switched-on, would

depend on the N -cutoff of the cavity, but it is well defined both mathematically and physically for any given setup.

Note that non-adiabatic switch-on of the ultrastrong coupling regime has been experimentally achieved (e.g., Ref. [34] of the novel version of the paper) and a theoretical analysis clearly shows the switch-on is performed with a local operation on the qubit (Ref. [65]).

Action. We added the following paragraph to the manuscript:

“Such a configuration can be prepared performing only local operations on the TLS, i.e., by non-adiabatically switching on its coupling to the cavity [34,35].”

2. ii) *In the introduction (3rd paragraph) the authors comment on the two main approximations in cavity QED, namely, the two level approximation and the single mode one. The latter is investigated in the ms. However, nothing is said on the TLS approximation. A naive argument says that some of these modes can be in resonance with higher transitions in the atom. I understand that this is model dependent but, at least, it must be discussed what it is the role of the TLS approximation on their results.*

Reply. The Reviewer raises a very good point, since nothing is said in the manuscript about the TLS approximation, which will certainly break down for high enough couplings. In order to verify whether our results are solid also for more realistic systems, we developed a generalised theory in which the qubit is described by a nonlinear system with many levels, and we have observed no qualitative difference in the physics, i.e., the field is still composed of a bound state and a free propagating wavefront. These results are shown in Fig. 1 below. We have addressed in the text that the TLS approximation might be compromised without any effect on the physics under discussion. Those novel results have been included in the new Supplemental Material.

Action. We have included a calculation with a general nonlinear system with many levels to the Supplemental Material, and the following paragraph to the manuscript:

“Finally, we stress that the phenomenology for the light field presented in this text is robust against the failure of the TLS approximation for the emitter. We have checked that, after substitution of the TLS by a cavity with a small Kerr nonlinearity, so that higher modes are also involved in the dynamics, the cavity light field in the ultrastrong coupling regime maintains the same features that we reported here.”

3. iii) *The authors cite several works [Their references 17, 23-29] where the multimode model was already discussed. Nature Comm. emphasizes novelty. Thus, they must clearly say what is novel in their work. What did the previous works discuss? Is the first time dealing with the dynamics?. Besides, two (at least) of these references report experiments. Are the effects described in the paper realizable in experiments?*

Reply. We thank the Reviewer for the constructive comment. Certainly, all these references recognize an hybridization with higher cavity modes due to larger couplings. The common point along all these works, as we stated in the text and the Reviewer themselves points out, is that they focus on spectral properties, and no dynamics is discussed. This miss the essential role that light propagation inside the cavity plays already in the regime of ultra-strong coupling. Our work is the first to evidence the failure of the single-mode Rabi model by making an exhaustive comparison of the dynamics as compared to a quasi-exact calculation for the full model. Dynamical experiments of the kind we propose are routinely performed in superconducting circuits, e.g., measuring Rabi oscillations in the strong coupling regime.

Figure 1: Dynamics of the field for $g/\omega_c = 0.6$, and the TLS substituted by a nonlinear cavity (adding a nonlinear term $U = \chi a^\dagger a^\dagger a a$ to the Hamiltonian) with $\chi = 10 \omega_c$.

Action. We have modified the following paragraph to emphasize the novelty of our results:

“However, in contrast to these previous works, our analysis reveals that the different nature of the eigenstates and their degeneracy have critical consequences on the system dynamics.”

4. iv) Below Eq. 14. They use only the acronym: in the PZW gauge.

Reply. We thank the referee for pointing this out.

Action. We have changed “PZW” to “Power-Zienau-Woolley”.

5. v) Below Eq. 14. What is the relation of ω_x with its bare value?

Reply. The exact form of the renormalised frequency depends on the system under investigation. In a bosonised model the matter part of the Hamiltonian is

$$H_M = \omega_b b^\dagger b + D(b + b^\dagger)^2,$$

where D is the coefficient of the P^2 term quadratic in the matter field, which in an electron gas would be linked to the plasma frequency. Performing a Bogoliubov rotation we can then diagonalise it in the form

$$H_M = \omega_x \tilde{b}^\dagger \tilde{b},$$

with the renormalised frequency $\omega_x = \sqrt{\omega_b^2 + 4\omega_b D}$. In the specific case of a two dimensional electron gas and of superconducting circuits the corrections can be found in the Refs. [36,37].

Action. We added a reference developing the PZW theory of a two dimensional electron gas, which reports the relation between the microscopic system’s parameters and the renormalised frequency ω_x .

6. [Maybe related to point i)] *The calculations are based on a very special initial condition. Do you have a protocol to prepare such a state in an experiment? What about other initial conditions?*

Reply. We already discussed briefly in the original manuscript two possible protocols to achieve the initial state we consider:

“This initial configuration is a superposition of excited states of the coupled light-matter system, which could be initialized by applying a pulse in a decoupled system and then rapidly turning on the coupling. As an alternative approach to obtain an initial excited configuration, one could also apply a suitable pulse to the coupled system in its ground state”

We believe that a further discussion on this point is unnecessary, since the main features we report on this work appear as long as the initial state is not the ground state. However, to disperse any doubt regarding this point, we have explicitly stated this on the new version of the manuscript.

Action. We added the following paragraph:

“In any case, the effects that we report here appear as long as the system is initially in some superposition of excited states.”

7. vii) *Their calculations use 50 modes [read in Fig. 2]. This introduces a cutoff of $50 \omega_c$. However, in a experiment the natural cutoff can be lower than $50 \omega_c$. What is the role of the cutoff (natural or numerical) in their results?*

Reply. We thank the Reviewer for the very relevant question. They are right in pointing out that a physical system might include a natural cutoff which may be different from the choice we made for our calculations. We have addressed the role of the cutoff N in the revised version of the text and in the Supplemental Material. In short, the results are very robust and do not depend significantly on the choice of N . Some features, like the temporal width of the revival peaks, do depend by N , but only logarithmically. We have now addressed this dependence explicitly in the text and the Supplemental Material. Notably, this more in-depth analysis have allowed us to give a more precise answer to some of the other questions brought up by the Reviewers, like the validity of the analytical approximations or the critical value of the coupling g at which the single mode fails.

Action. We have addressed in detail the role of the cutoff N on the revised version of the manuscript (in the text between Eqs. (9) and (11), approximately) and on a new Supplemental Material.

8. viii) *For me it was surprising how localized are the wave packets travelling in the cavity. In principle, the atom is punctual and the emitting field should be a delocalized wave packet. Their calculations say that my guess is completely wrong, but I do not see where. Can the authors help me?*

A punctual emitter does not emit a delocalized wave packet. Already in the weak coupling regime, one can describe, within the Wigner-Weisskopf approximation, the emission of a photonic wave packet of size c/Γ , where Γ is the inverse life-time of the emitter. Even at this basic level of description one obtains a wavepacket that becomes more localized as the coupling rate to the continuum of modes increases.

9. ix) *The authors explain the field "cloud" around the TLS by computing the ground state. It would be also interesting to compute the $\langle \sigma^\dagger \sigma \rangle$ in the ground state to explain the $1/2$ between revivals [Cf. fig*

2b]. They obtain this 1/2 in Eq. (10) (without referring to Fig.2b). This analytical calculation was done in the limit $\omega_x=0$.

Reply. The referee is right in pointing this out. The ground state of the system indeed gives $\langle \sigma^\dagger \sigma \rangle = 1/2$, which matches the value observed between the revivals.

Action. We have included a reference to Fig. 2b after Eq. (11) and the following paragraph to emphasize the relaxation to a local ground state:

“The results obtained confirm that the ground state of a TLS non-perturbatively coupled to a cavity is indeed constituted by a localized cloud of photons around the TLS, which is in a superposition with a population corresponding to that observed in the revivals $\langle n_\sigma \rangle = 1/2$ ”.

10. x) [Maybe related to viii)] I miss a physical picture explaining that adding modes the revivals and wave packet evolution become sharper.

Reply. By adding more modes, the overlap between the two distinct cavity states associated to the TLS states $|\pm\rangle$ becomes smaller, i.e., adding modes makes the two possible cavity states diverge faster from each other. Mathematically, this can be understood by noticing that the photonic modes of the cavity we consider are harmonics with decreasing wavelengths. The field in the cavity is a superposition of those harmonics, and its sharpness is thus limited by the shortest wavelength of the coupled harmonics. This fast collapse (and revival) of the overlap is directly related to the dynamical features that we report. We discussed this point in more details in a section of the Supplemental Material. Interestingly, these sharp collapse and revivals are very much related to previous works on the dynamical description of wave-function collapse as a unitary evolution.

Action. These questions have been discussed in the new version of the manuscript (between equations (9) and (11)) and in the novel Supplemental Material.

xi) In the last figure (4b), the eigenvalues using the single mode and the multimode deviate already at $g=0.2$. Besides, the authors write at the end of the 5th paragraph, “.. a low-energy spectrum of the single mode description that does not differ greatly from the full, multi-mode case, if one uses distinct fitting parameters...”. If the experiments are fitted with a single mode, the coupling is over or under estimated?

Reply. In the example mentioned by the referee ($g/\omega_c = 0.2$, $\omega_x = 1$), the coupling and qubit frequency that one obtains by fitting with a single-mode is $g/\omega_c = 0.194$ and $\omega_x = 0.87$. As another example, for the case of $g/\omega_c = 0.4$, we get $g = 0.38$ and $\omega_x = 0.58$. For $g/\omega_c = 0.6$, we obtain $g = 0.602$ and $\omega_x = 0.18$. In general, one observes a tendency to underestimate much more the value of ω_x rather than g . The situation may be different in setups with further fitting parameters, such in superconducting circuits, where we can have tunnel splitting and superconducting bias. A detailed study of this issue will be the matter of a future work.

Reviewer #2

The study is very thorough and well-explained. It is also extremely timely in view of the recently strongly revived interest in the quantum Rabi model. It shows that a model that is commonly used to describe cavity QED has to be taken with great care when reaching the aspired regimes of ultrastrong and deep strong coupling, and it discusses the deviating physics that the more realistic multi-mode model

presents. The article will thus be of high interest to many physicists and I recommend its publication in *Nature Communications*.

Before that, however, I would like the authors to address a couple of issues (many of them only cosmetic), which I outline below.

Reply. We are very grateful to the Reviewer for his/her positive appreciation of the manuscript.

1. As indicated above, I find the strong focus on the superluminal signaling in title and abstract unnecessary and somewhat exaggerated, since it is not really surprising in such an effective single-mode approximation. Shifting the focus more strongly on the relevant parts of how a correct description is obtained and what its physical consequences are could improve the article. But this is maybe a matter of taste.

Reply. We thank the Reviewer for the suggestion. Although we hear the Reviewer's opinion, we believe that the link between superluminal signalling and the in USC regime to be an original result. This Reviewer and Reviewer #4 both suggested us to change our title, but it seems to us their suggestions go in opposite directions. In order to avoid making our title in any way misleading, we thus removed any mention of superluminal signalling from the new title: "Effect of light propagation in the non-perturbative regime of cavity quantum electrodynamics". Note that we have removed an explicit mention to the Rabi model, since the new version of the work shows that the effects of light propagation appear also in more general models.

Action The title has been changed to "Effect of light propagation in the non-perturbative regime of cavity quantum electrodynamics".

2. On p. 1, one finds Unless $\tau_R \gg L/2c$. . . Is not a simple $>$ already sufficient?

Reply. In a single-mode approximation, the very concept of propagation is absent, and the probability for the observer to measure a photon is an analytic function of both time and coupling strength. As such there is no sharp transition between a "causality respecting" and a "causality violating" situations. For t sizeably smaller than τ_R the probability of observing a photon and thus transmit superluminal information is vanishing, in the opposite case the probability is close to one. Since τ_R is here considered as a time-scale rather than a specific point in time, we believe that the symbol \gg here is thus more appropriate.

Action. We have slightly rephrased the text to better identify τ_R as a timescale rather than a given point in time:

"After a timescale $\tau_R \approx 2\pi g^{-1}$..."

3. In Eq. (4) and below, the symbol N is not properly introduced. Implicitly, it becomes clear that it is the number of modes, but that should be mentioned as well as why the limit $N \rightarrow \infty$ is taken.

Reply. The referee is right, and we thank them for making us aware of this error. Regarding the limit $N \rightarrow \infty$, this matter has been treated with greater care in the new version of the manuscript, and such a limit is not taken anymore. The results are now discussed considering N as another physical parameter. The main changes on the new version of the manuscript and also the new Supplemental Material are devoted to study the effect of keeping a finite N .

Action. We have introduced the meaning of the symbol N . The new version of the manuscript and the Supplemental Material deals in detail with the limit $N \rightarrow \infty$, and the influence of not taking such a

limit in our results.

4. *Even if well-known, the acronym PZW gauge should be defined.*

Reply. The referee is right and this acronym has been defined, also as a response to Reviewer #1's comment #4.

Action. The acronym has been defined.

5. *In Fig. 2c [sic], the peaks above the dashed line are hardly visible, at least on my computer screen.*

Action. We have considered the Reviewer's comment and included an inset in the same figure, zooming around the revival lines to make them clearly visible. The caption has been updated accordingly.

6. *Again in Fig. 2c, in the multimode case there are vertical stripes, visible even at very low g . Are these numerical artifacts or are they physical?*

Reply. These vertical lines are physical; they correspond to distortions on the Rabi oscillations occurring at times multiple of the cavity roundtrip time and induced by the free-propagating wavepackets, which preclude the revival peaks emerging at higher values of g . The fact that these lines are not artifacts is better appreciated in the new insets of Figure 2, and also in Figure S3 in the Supplemental Material.

Action. We have included in the text the following explicit mention to these distortions to clarify their origin.

“Even for lower values of the coupling—before these revival peaks are fully formed—one can observe a perturbation of the Rabi oscillations taking place at those specific times.”

7. *Figure 4a shows the two contributions of the localized field and the propagating wave. What is the exact ratio between the amplitudes of both? From Eq. (12), derived analytic in the approximate model, it appears that it is a universal number, but in the numerics does it depend on the actual model parameters?*

Reply. The changes included in the text now clarify that the model gives actually a very accurate description of the dynamics once the coupling is big enough to enter the regime of collapse and revivals. Therefore, the conclusions obtained from the numerics are no different to those drawn from the model. Answering to the question, the number thus depends only on the initial state. This has been explicitly stated in the new version of the text.

Action. We have added the following extract to clarify the point:

“(…), and the ratio between them will depend on the initial state (being 1/2 in our particular case)”.

8. *Around Fig. 4b, a discussion of the degeneracies of the levels at $g \rightarrow 0$ and $g \rightarrow \infty$ seems in order.*

Reply. We agree with the Reviewer that this is an interesting point worth discussing. However, we find that this feature is not unique from the multi-mode description, and has been analyzed by other authors before. The references originally provided in the paragraph

“In the large-coupling limit, both models feature a series of equispaced energy levels similar to the bare ones, a result shown above in the derivation of H'_I and well known for the single-mode case [20,28,62,63]”

already offer an insightful analysis of these degeneracies. Therefore, we believe that further elaboration on this feature is, albeit interesting in the context of multi-mode dynamics, out of the scope of the present manuscript.

9. *I am missing two references on experiments on the Rabi model:*

1. *Experimentally simulating the dynamics of quantum light and matter at deep-strong coupling. N. K. Langford, et al., Nature Communications 8, Article number: 1715 (2017)*
2. *Analog quantum simulation of the Rabi model in the ultra-strong coupling regime J. Braumüller, et al., Nature Communications 8, Article number: 779 (2017)*

These works realize the single-mode Rabi model in an effective way by alternating between Jaynes-Cummings and Anti-Jaynes-Cummings terms. In how far do the present findings apply to such synthetic models?

Reply. We thank the Reviewer for bringing these works into our attention. We believe that these proposals are indeed successful in obtaining synthetic single-mode Rabi models, and therefore, our findings do not apply here. This can be understood since these methods do not actually increase the coupling (which might induce the hybridization with higher cavity modes) but reduce effectively the frequency of the only mode coupled to the TLS by applying suitable drivings and changing the rotating frame. The works are, however, relevant and worth mentioning in the text.

Action. References to synthetic implementations of the Rabi model have been added to the manuscript, in the paragraph:

“Its mathematical properties [23] and the possible implementations with synthetic models [24,25] have also become object of interest.”

10. *In the methods, the formula for the Pochhammer symbols in terms of a_i is confusing, since the symbols a_n are already used for the annihilation operators of the photon modes.*

Reply. We thank the Reviewer for the comment. This has hopefully been clarified in the new version.

Action. We have substituted the symbol a_n by $(z)_n$ in order to avoid confusion with the annihilation operators.

Reviewer #3

I think this is a really interesting and solid piece of work, reporting several important results. In particular, I believe that the argument of causality violation is specially relevant and timely, given the current state of the field, reporting ever-higher values of the coupling. The single-mode Rabi model has been used as a very standard tool to describe and analyze these systems in the ultra-strong coupling, and here the authors show unambiguously (see for instance the difference between single-mode and multi-mode behaviors in Fig. 1(a-b)) that something very important is missing in the single-mode description. I

believe that it is most likely that works like the present one will help improving the quality and quantity of future publications in the field of cavity QED.

Reply. We are most thankful to the Reviewer for his/her positive feedback.

1. Several experimental works (see for instance Bayer et al. Nano Lett., 17, 6340 (2017), or Yoshihara Nat. Phys., 13, 44, (2017)) have already reported values of the light-matter coupling rate well above $g/w = 0.6$, which is approximately where the authors show that the single-mode Rabi model is no longer valid. The authors should comment on the link between their work and these experimental results, and address to which extent are those results affected by the ones reported here.

Reply. As we show in a clear way in the new version of the text, the extent to which those works are affected by the results we report is very much influenced by the number of modes involved in each system. For instance, the system used in Yoshihara *et al.* is de facto a real single-mode Rabi model, and will not be affected by the hybridization with higher cavity modes. In the case of the work by Bayer *et al.* instead, the higher cavity modes play a very important role for the larger values of the coupling, where the second order lower polariton mode is pushed in the spectral range between the lower and the upper polariton (Fig. 3 of the Nano Letters). A direct application of our quantitative results to the system studied by Bayer *et al.* is not possible, due to the very different setup involved. The resonator used by Bayer *et al.* is in fact strongly non-harmonic, while the matter resonance is harmonic. Still what our results show is that, in order to study the dynamics of such a system, propagation effects cannot be neglected.

2. In this work, it is claimed that the well-known virtual photon population of the ground state in the ultra strong coupling regime (which is known to yield no emission) takes the form of a photonic cloud localized around the two-level system. I think the authors should comment on how does the spatial extension of this state depends on the coupling rate, and how could this observation be measured experimentally (given that these photons yield no observable emission)

Reply. In the new version, we have clarified in which range of coupling rates does the system enter a regime well described by our analytical equations, that disregard the $\omega_x \sigma_z / 2$ term in the Hamiltonian. It is in that regime that the localized bound state manifests clearly. Interestingly, in this regime, the profile of this bound cloud is independent of g , but only depends on the number of modes N involved in the dynamics. This conclusion is completely analogous to the fact that the temporal profile of the revival peaks is as well independent from g .

Action. We clarified these matters in the main text and in the new Supplemental Material.

3. The color codes using in the manuscript are confusing sometimes. In particular, in Fig. 2(b), red is used to represent the multi-mode description, and blue for the single-mode. In Fig. 4(b), it is the other way around. I think this should be fixed for the sake of consistency.

Reply. We thank the referee for noticing the inconsistency.

Action. The color code has been swapped in Fig. 4(b) in order to avoid confusion.

Reviewer #4

After a brief research, some previous works mentioned that multimode should be considered when the coupling is strong enough, (such as Sundaresan, Neereja M., et al. "Beyond strong coupling in a multimode cavity." Physical Review X5.2 (2015): 021035) but I did not find any article that showed single mode QRM would lead to superluminal signals. In my opinion, this paper has solid results and strong arguments while still needs to be revised and resubmitted. Following are some suggestions/questions and some mistakes I found.

Reply. We thank the Reviewer for his/her time and consideration and positive opinion of the paper. We are happy to consider his/her suggestions and comments.

1. The title is somehow misleading. I think it should convey the idea that multi-mode QRM is better since it resolves the problem of unphysical superluminal signal that caused by single-mode QRM. Thinking about something like: "Resolution of Superluminal Signalling: Multi-mode Quantum Rabi Model".

Reply. As explained in the reply to Reviewer #2, following the comments from both Reviewers #2 and #4, and in an attempt to avoid any risk of our title to be misleading, we have changed it to "Effect of light propagation in the non-perturbative regime of cavity quantum electrodynamics". We hope it will convey in a clearer way the content of the manuscript.

Action. The title has been changed to 'Effect of light propagation in the non-perturbative regime of cavity quantum electrodynamics'.

2. I believe that the Hamiltonian (4) can be easily generalized to some other kinds of systems such as circuit-QED, where the coupling strength g_n to different modes are engineerable.

Reply. This Reviewer is correct in noticing that Hamiltonian (4) can be easily generalised describe arbitrary superconducting circuits (or even dielectric) configurations, but the values of the g_n will still be self-consistently linked with the mode structure of the resonator. In general any microscopic, multi-mode model will respect causality, and this poses constraints on the value of the g_n once the resonator geometry and modal structure is fixed. As an example, in Ref. [38] in the new version of the manuscript, various circuit configurations are considered, clearly demonstrating how the specific kind of coupling limits the values of the couplings (and thus the possibility to achieve a phase transition).

Action. We have enriched the discussion about other kinds of system adding Reference [38].

3. If only a finite number of modes are considered in equation (4), my nave guess is that we can re-diagonalize the H matrix (4) to find only one superposition of photon modes coupled to the TLS, which might be able to reduce equation (4) back into equation (1).

Reply. We believe that the diagonalization proposed by the Reviewer is not possible. Although the matter system is coupled to a well defined superposition of photonic modes, those modes all have different bare energies. As such if we attempt the kind of diagonalization this Reviewer is suggesting, we would end up with the matter mode coupled to a single photonic mode, but itself coupled to other mutually interacting $N - 1$ photonic modes. One way to check this is to attempt such a procedure in the simplest case of a Jaynes Cummings Hamiltonian.

4. Although the analytical result resembles the numerical solution, it might still need an analytical

explanation: why is H_{II} neglectable compared with H_I ?

Reply. We thank the Reviewer for the insightful question. We believe this point has been greatly clarified in the new version of the manuscript; the reason for H_{II} being negligible is due to the fact the overlap between the cavity states associated to each of the two qubit states that this term can couple tends to zero as the number of modes is increased. This question is now discussed in depth both in the text and in a new Supplemental Material.

Action. We have added an extensive discussion on the validity of the analytical results and the negligible role of H_{II} in the new version of the text and in the new Supplemental Material.

5. One main reason we are still using single-mode QRM is that driving and damping can be easily added in, can we extend it and add the driving and damping to this multi-mode picture?

Reply. Driving can be readily added to our system as an extra term in the Hamiltonian. Regarding losses, there is no reason, from a formal point of view, not to treat them in the multi-mode picture in the same way they are treated in the single-mode case. Regarding our numerical methods to approach the problem, we believe that some techniques of the field open quantum systems, such as the method of quantum trajectories, can be readily implemented with the technique of Matrix Product States that we have used here.

Action. We have added a comment regarding losses:

Those virtual photons have been demonstrated to exist also in lossy systems [52], although once the coupling with the environment is properly considered [35,53-56] their non-radiative nature becomes apparent. Our results provide a more transparent way to understand them as a localized, bound state of photons; in future works, the methods that we present here might be applied to study their properties in lossy systems.

6. The conclusion needs to be clearer: at what g/ω_c regime the single-mode QRM becomes invalid? (0.6? 0.2? or it depends?)

Reply. Again, we thank the Reviewer for the very relevant question, which we believe has been successfully addressed in the new version of the text. The answer is, it depends on the value of N , but only logarithmically. In our new version of the text, we are not taking the limit $N \rightarrow \infty$ anymore, but keeping N as an extra parameter. We have obtained that, for the realistic range of $N \in [10, 100]$ considered in the text, this critical value of the coupling is $g/\omega_c \approx 0.25$. We refer the Reviewer to the new version of the text, and the new Supplemental Material, for an in-depth analysis of this question.

Action. We have included in the Supplementa Material an analysis of the critical g/ω_c at which the single-mode QRM becomes invalid, and we have discussed these results in the main text.

7. Is there a superluminal signal when coupling strength is small (not in ultrastrong regime)? If there is, why it is not a problem. If not, why it emerges after a certain threshold.

Reply. We refer this Reviewer to the answer we gave above to a similar question from Reviewer #2. Essentially there is not a sharp threshold between two causality violating / not violating regimes. Although in a single mode approximation it would seem that we still have causality violation because the observer has access to a non-zero photonic field for any time $t > 0$, the intensity of this field vanishes for $t \ll \tau_R$. As long as the timescale for the formation of a delocalized state remains larger than the timescale of light propagation, there is thus no problem with causality.

Action. We have stressed in the text that there is not a sharp threshold by explicitly talking in terms of *timescales* (c.f. response to Reviewer #2).

Typos:

1. In the captions of Fig. 2(c) and Fig. 4, I think ω_a should be ω_c .
2. In Equation (6), $|\psi(t)\rangle$ should be $|\psi(t)\rangle_{\pm}$ as it represents two different states.
3. On page 4, ... seems to clearly attribute the regular peaks in Fig. 2 with period..., Fig. 2 should be specified as Fig. 2(b).

Reply. We thank the Reviewer for noticing these misprints.

Action. All these typos have been corrected in the new version.

Summary of most relevant changes

- Updated discussion on the role of the total number of modes involved, N , the validity of the analytical equations, and the values of coupling rates at which the single-mode Rabi model breaks down. Results presented both in the main text and, with more detail, in a new Supplemental Material.
- Fig. 2 has been updated with an inset depicting the revivals.
- Fig. 4(a) includes now a comparison between numerical and analytical results. Colors in Fig. 4(b) has been swapped for consistency.
- Discussion on mode general models besides the Rabi model, including calculations with a non-linear cavity instead of a two-level system, included in the Supplemental Material.
- Typos corrected, and small additions to address specific Reviewers' comments.

REVIEWERS' COMMENTS:

Reviewer #1 (Remarks to the Author):

The authors have tackled all the points that I raised. In fact they went thoroughly, which I acknowledge.

In my opinion, the paper is ready to be published in Nat. Comms.

Reviewer #2 (Remarks to the Author):

The authors have addressed the rather exhaustive list of concerns and comments by myself as well as the other Referees. I very much appreciate the extreme thoroughness of their response, which has added several important discussion and put the manuscript on an even broader footing.

Even more so than before, I recommend publication of this manuscript; this time without further revisions necessary.

Two small remarks:

1. Regarding the title, the authors write they feel the suggestions from Referee #4 and me go into different directions. Actually, I believe both of us had something similar in mind. The previous title suggested that the focus was on the superluminal signaling, while the actual content of the manuscript is how this superluminal signaling becomes unphysical once a correct description is chosen. From this point of view, I did find the suggestion by the Referee #4 rather fitting.

2. A typo: "an the ratio between them will depend"

Reviewer #3 (Remarks to the Author):

I have thoroughly reviewed the revised manuscript, and I found the revised version very satisfactory. It has not only addressed my questions adequately, but also provide very complete answers to the questions raised by other referees.

Again, I emphasise that this is a very timely research topic and I expect the paper will make a great impact to the research areas of quantum dissipative systems, cavity QED and quantum information processing.

Therefore, I recommend its publication in Nature Comm. in this version without further delay.

Reviewer #4 (Remarks to the Author):

In this second version, I believe that most of my questions are well answered, and I only have a few comments to address, which should be fixed easily. I recommend Nature Communication to publish this paper after a slight modification.

1. Page 5, "an the ratio" should be "and the ratio".
2. I do enjoy this new supplemental material part especially Equation (S2). But I am not sure if the result will converge when N goes to infinity (The convergence of Eqn.S2 might not be sufficient), and it might be hard to prove. For the sake of safety, the author may need to be more careful when N is larger than 100.
3. Maybe the methods part should be included in the supplemental material, but it's only a matter of style.

Reviewer #2

1. Regarding the title, the authors write they feel the suggestions from Referee #4 and me go into different directions. Actually, I believe both of us had something similar in mind. The previous title suggested that the focus was on the superluminal signaling, while the actual content of the manuscript is how this superluminal signaling becomes unphysical once a correct description is chosen. From this point of view, I did find the suggestion by the Referee #4 rather fitting.

We thank the Reviewer for the clarification. In the light of their comment, and given that both Reviewers agree, we have decided to revise again the title, following closely the suggestion by Reviewer #4, who proposed the title “Resolution of Superluminal Signalling: Multi-mode Quantum Rabi Model”. Our only modification to this suggestion is that we prefer not to speak specifically of the Rabi Model, since in the new version of the manuscript (posterior to Reviewer #4 original suggestion) we prove that the effects that we report are general and manifest in other multi-mode models. Therefore, we have chosen the title “Resolution of superluminal signalling in non-perturbative cavity quantum electrodynamics”.

2. A typo: “an the ratio between them will depend”

We thank the Reviewer for noticing this typo. The mistake has been amended.

Reviewer #4

1. Page 5, “an the ratio” should be “and the ratio”.

We thank the Reviewer for pointing this out. We have corrected this typo.

2. I do enjoy this new supplemental material part especially Equation (S2). But I am not sure if the result will converge when N goes to infinity (The convergence of Eqn.S2 might not be sufficient), and it might be hard to prove. For the sake of safety, the author may need to be more careful when N is larger than 100.

The key physical quantity on which the value of N has an impact is the overlap O_N . Such a quantity is a monotonically decreasing function of N , converging to zero as $N \rightarrow \infty$. In consequence, all related quantities present in our theoretical treatment, as g_c in Supplemental Equation (2), are well behaved for arbitrarily large N .

3. Maybe the methods part should be included in the supplemental material, but it's only a matter of style.

We thank the Reviewer for the suggestion. We agree this is a matter of style; however, we will choose to keep the Methods part as it is, since we believe it provides the type of technical information that is more fitting for this section.

Summary of most relevant changes

- The title has been changed to “Resolution of superluminal signalling in non-perturbative cavity quantum electrodynamics”, according to the suggestions from Reviewers #2 and #4 and taking into account that our latest version of the manuscript includes general results beyond the Rabi model.
- Typos have been corrected.
- Although no Reviewer pointed this out, we realised that through the text we referred to states localised close to the matter system as either “bound” or “bounded”. We uniformed the notation using the correct “bound” form.